# Split Application under Reduced Nitrogen Rate Favors High Yield by Altering Endogenous Hormones and C/N Ratio in Sweet Potato

**Xiangbei Du [1], Xinyue Zhang [2], Lingcong Kong [1,\*] and Min Xi [3,\*]**

[1] Crop Research Institute, Anhui Academy of Agricultural Sciences, Hefei 230031, China; duxiangbei@aaas.org.cn

[2] Jiaxing Academy of Agricultural Sciences, Jiaxing 314016, China; drzhangxinyue@163.com

[3] Rice Research Institute, Anhui Academy of Agricultural Sciences, Hefei 230031, China

\* Correspondence: anhuiganshu@126.com or konglingcong@aaas.org.cn (L.K.); min8552@126.com or ximin@aaas.org.cn (M.X.); Tel.: +86-551-65149831 (L.K.); +86-551-65121322 (M.X.)

**Abstract:** A process for reducing the nitrogen (N) application rate while maintaining sweet potato yield urgently needs to be determined. A two-year pot experiment was conducted with three N management strategies to explore the mechanism underlying yield increase caused by a split application under a reduced N rate through an investigation of the changes in the carbon (C)-N metabolism and endogenous hormone. Results revealed that, compared with conventional basal N management, split application under a reduced N rate increased storage root yield by 22.1% through improving the storage root number and mean storage root weight by 12.3% and 10.2%, respectively. During the storage root formation period, split application under a reduced N rate decreased the soil-available N (AV-N) content and N content in storage root, inducing elevated C content, C/N ratio, auxin (IAA) content, zeatin and zeatin riboside (Z + ZR) content and reduced abscisic acid (ABA) content in storage roots, promoting storage root formation. During the storage root bulking period, split application under a reduced N rate appropriately elevated the soil AV-N content and N content in the storage root which, together with increased ABA content, which enhanced C content and C/N ratio in the storage root, resulted in an improved mean storage root weight. These results will facilitate the generation of appropriate N management strategies to improve sweet potato productivity.

**Keywords:** sweet potato (*Ipomoea batatas* (L.) Lam); nitrogen management strategies; soil-available nitrogen content; carbon–nitrogen metabolism; endogenous hormones

## 1. Introduction

Sweet potato (*Ipomoea batatas* (L.) Lam) is an important staple food in tropical, subtropical and temperate regions and positioned as the seventh most important crop in the world [1]. As such, it is essential to secure the continuous stability and improvement of sweet potato production. As an important root crop, sweet potato yield is determined by the storage root number and mean storage root weight [2], which are closely related to the initial formation and development of storage root.

Nitrogen (N) management is an important agronomic practice affecting the initial formation and development of storage root [3]. N fertilizer (Urea) is often excessively applied (90–200 kg ha$^{-1}$) as basal fertilizer in sweet potato production in China, which is far beyond the amount of recommended N application (75–135 kg ha$^{-1}$) [4]. Although the mean storage root weight could be enhanced by increasing the N application due to higher leaf source strength (e.g., leaf area index, leaf area duration, net assimilation rate) during the storage root bulking period [5], increased N application tends to hinder the formation of storage roots during the storage root formation period [6]. Therefore, excessive

N does not increase sweet potato yields and sometimes even decreases sweet potato yields. Moreover, excessive use of chemical N fertilizer also causes serious environmental pollution, such as groundwater contamination and soil acidification [7]. Hence, the N application rate needs to be reduced while maintaining sweet potato yield. However, reduced N is commonly applied as basal fertilization once, which decreases soil-available N (AV-N) during the whole plant growth stage, resulting in early leaf senescence and decreased photoassimilation and photosynthate supplementation during the middle and later growth stages [4,8]. In contrast, split N application is a recommended way to use N efficiently by specifically synchronizing N supply with a plant's ability to utilize nutrients [9]. In summary, we hypothesize that a split application of reduced N may be beneficial to both the initial formation and development of storage root which may then enhance sweet potato yield by increasing both the storage root number and mean storage root weight.

Sweet potato adventitious roots form in the early stage of root development, and then, some adventitious roots thicken and develop into storage roots. N regulates plant growth and development in two ways: (a) regulating carbon (C)-N metabolism and (b) altering hormonal signals [10]. C-N metabolism is a fundamental process in crops and closely associated with the synthesis and transformation of photosynthetic product and protein. C content, N content and C/N ratio are usually used to indicate the intensity of C-N metabolism in crops [11]. Endogenous hormones, including auxin (IAA), zeatin and zeatin riboside (Z + ZR), and abscisic acid (ABA), play important roles in the formation and development of storage root [12–15]. Moreover, the same endogenous hormone may differentially affect different stages of storage root development. Many studies have focused on the effect of N application rates on the C-N metabolism and hormonal signals of sweet potato [16,17], soybean [18], maize [19] and wheat [20] and have demonstrated that N regulates the accumulation, transportation and unloading of photoassimilate at several levels of integration through the control of hormone content and proportion. However, to date, knowledge on the effect of a split application of reduced N on the change characteristics of C-N metabolism and endogenous hormone and their relationships with the formation and development of storage root in sweet potato is limited.

The experiments were carried out to analyze the physiological mechanisms of split N application effects on sweet potato yield under reduced N input. The objectives of this study were to identify the changes in C-N metabolism and endogenous hormones during the storage root formation period and storage root bulking period under different N management strategies and to investigate their relationships with the formation and development of storage root in sweet potato. This study will help us understand the response of sweet potato yield to a split application of reduced N and facilitate improvements in sweet potato productivity through appropriate N management.

## 2. Materials and Methods

### 2.1. Experimental Site and Plant Material

Two-year pot experiments were conducted in a simple plastic shelter in 2016 and 2017 at Hefei, Anhui, China (31°89′ N, 117°25′ E). Shangshu 19, a widely cultivated sweet potato cultivar that tends to exhibit early storage root formation, was selected for the pot experiment.

### 2.2. Experimental Design and Management

Three treatments were carried out in the experiment: the conventional N management treatment, in which 2.0 g N pot$^{-1}$ was applied as basal fertilizer once (CM) and two 20.0% reduced N fertilization treatments with a total N value of 1.6 g pot$^{-1}$, where one treatment involved 100.0% N applied as basal fertilizer (RB), and the other treatment involved a split application of 50.0% N at transplanting and 50.0% N at 35 days after transplanting (DAT) (RS). N fertilizer was applied in the form of urea. A recommended dose of 1.8 g P pot$^{-1}$ (triple superphosphate) and 3.0 g K pot$^{-1}$ (potassium sulfate) was applied as basal fertilizer to each pot. The applied fertilizer in the pot experiment was equivalent

to 100 and 80 kg N ha$^{-1}$, 90 kg P ha$^{-1}$ and 150 kg K ha$^{-1}$ in the field experiment. The N–P–K fertilizer was applied as a liquid.

The pot experiment was arranged in a complete randomized design with three replications. Each replication of treatment consisted of 50 pots, which were arranged in 2 rows with 25 pots in each row. The pots were next to each other in each replication of treatment, but there was 1 m spacing between two treatments. The pots (60 cm in diameter and 55 cm high) were filled with 30 kg of well-mixed soil. The pot soil was yellow brown loam with a pH of 5.85, containing 11.60 g kg$^{-1}$ organic matter, 0.82 g kg$^{-1}$ total N, 9.36 mg kg$^{-1}$ AV-N, 34.15 mg kg$^{-1}$ available phosphorus (P) and 69.35 mg kg$^{-1}$ available potassium (K). Individual healthy and uniform seedlings were transplanted into plastic pots on 29 May and 1 Jun in 2016 and 2017, respectively, with one plant in each pot. Tap water was used for irrigation when plants needed to be watered. Other agronomic practices were uniformly managed following the local agricultural practices.

### 2.3. Data Collection

#### 2.3.1. Soil-Available Nitrogen

Soil at a depth of 0–20 cm was collected from each pot at 35 and 80 DAT. Moreover, the soil of 35 DAT was sampled before N application. An aqueous extraction of 5 g fresh and uniformly mixed soil samples in 0.01 M CaCl$_2$ was used to determine the soil nitrate-nitrogen (NO$_3$-N) and ammonium nitrogen (NH$_4$-N) levels. The levels of NO$_3$-N and NH$_4$-N were measured by a continuous flow analyzer (Bran and Luebbe TRAACS Model 2000 Analyzer, BRAN+LUEBBE, Hamburg, Germany) [21]. The AV-N content was calculated as the sum of the NO$_3$-N and NH$_4$-N contents.

#### 2.3.2. Sweet Potato Yield and Yield Component

Ten sweet potato plants were harvested by hand in each pot at 115 DAT. The yield components (storage root (diameter > 2.0 cm) number and mean storage root weight) and sweet potato yields were determined based on the data of ten random pots in each treatment.

#### 2.3.3. Nitrogen Content, Carbon Content and Endogenous Hormones

The number of storage roots was essentially stable at 35 DAT [22,23]. Storage roots with diameters greater than 5 mm were sampled at 35 and 80 DAT according to Wang et al. [24]. The storage root skin was peeled, and the storage parenchyma was split in half, and then divided into two groups and cut into slices. Group slices were first heated at 105 °C for 30 min and subsequently dried to a constant weight at 65 °C. The dry slices were pulverized into a powder and then sieved through a 120-mesh sieve for N and C analysis. The other group of slices were immediately frozen in liquid N and then stored at −80 °C until needed for the determination of endogenous hormone content. The measurements were repeated three times independently, with at least three storage roots for each treatment.

N content was determined according to the Kjeldahl method [25]. The C content of storage roots was also analyzed by an elemental analyzer (Thermo Fisher Scientific, Bremen, Germany).

Hormone contents were determined by high-performance liquid chromatography (HPLC). The details of the ABA, IAA and Z + ZR hormone extraction, purification and determination have been described by Yang et al. [26] and Yang et al. [27]. One gram of storage root was ground with a mortar under ice bath conditions, mixed with 80.0% cold methanol, wrapped well in plastic and refrigerated overnight at 4 °C. The extract was filtered. The mortar was rinsed twice with 10 mL methanol, and the solution was filtered and mixed with the filtered extract. Then, the methanol of the mixture was removed by evaporation under reduced pressure at 40 °C. The remaining water phase was completely transferred to the triangle and extracted twice with petroleum ether, and then, the organic layer was discarded. The pH of the remaining water phase was adjusted to 6.4, and 0.5 g polyvinylpyrrolidone was added. Furthermore, the solution was filtered after the 30 min ultrasound treatment. The pH of the filtrate was adjusted to 2.9 and extracted three times with ethyl acetate. The organic layer was

retained and dried by evaporation under reduced pressure. The obtained residue was dissolved in 2 mL, and the solution was filtered by 0.45 μm microporous membrane filtration. The obtained solution was used to measure the hormones. Analyses were carried out using a Waters Alliance 2412 HPLC system (Waters, MA, USA). A 20 μL aliquot of that solution was injected into a fixed 20 μL loop for loading onto a $C_{18}$ reverse-phase column (4.6 × 150 mm, 5 μm particle size). Samples were eluted from the column by a Waters series 515 pump at 25 °C with a flow rate of 0.6 mL min$^{-1}$. Hormone peaks were detected by a photodiode array detector (Waters 2998 Separations Module, Waters, MA, USA) with an absorbance at 254 nm.

*2.4. Statistical Analysis*

Data were subjected to ANOVA to determine the difference between the treatment means using SPSS 20.0. The means were tested with the least significant difference (LSD) test, and the significance level was set at the 0.05 probability level. Microsoft Excel 2017 (Microsoft Corporation, Redmond, WA, USA) and Origin software (Version 2018, OriginPro, 2018) were used for data visualization.

## 3. Results

*3.1. Soil-Available N Content*

The soil N availability significantly decreased during the storage root formation period but increased during the storage root bulking period in the RS treatment compared with that in the CM and RB treatments. The AV-N content in the CM treatment was the highest, followed by that in the RB treatment, and RS treatment had the lowest AV-N content at 35 DAT. However, RS treatment recorded the highest AV-N content due to the split application of 50% N during 35 DAT, followed by CM treatment, and RB treatment had the lowest AV-N content at 80 DAT (Figure 1).

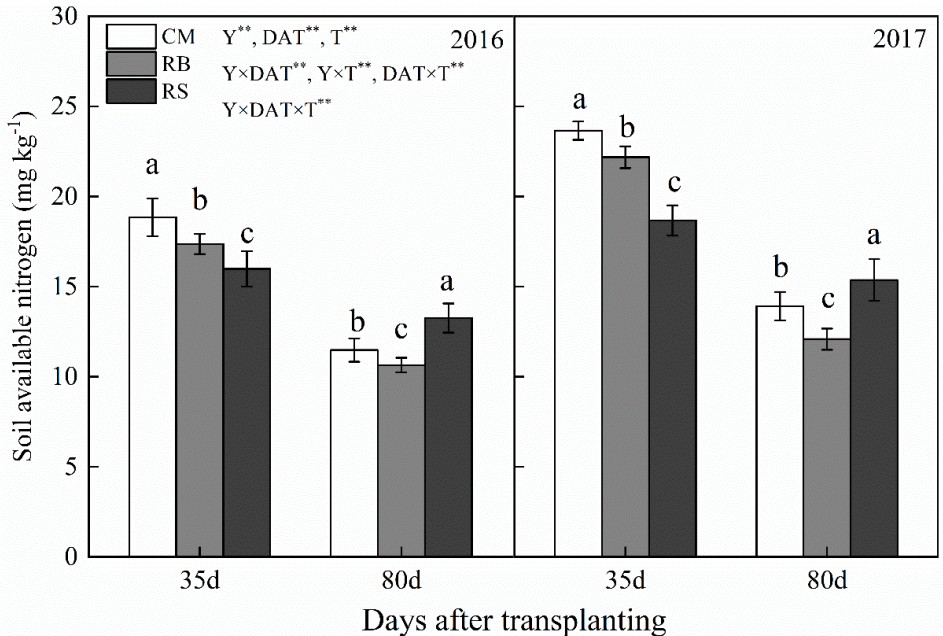

**Figure 1.** Soil AV-N levels of three N management strategies at soil depths of 0–20 cm in 2016 and 2017. Note: CM: conventional N management; RB: 20.0% reduced N was applied as basal; RS: split application of 20.0% reduced N at transplanting and 35 DAT. Y, year; DAT, days after transplanting; T, treatment. Values are the treatment means for a given management strategy within a particular year ($n = 3$). Error bars represent the standard error of the mean. ** indicate significance at $p < 0.01$ probability level. The different lowercase letters in the same column indicate significant differences at $p \leq 0.05$.

### 3.2. Sweet Potato Yield and Yield Components

The sweet potato yield and yield components were significantly affected by the N management strategies (Figure 2). Sweet potato yield significantly decreased under the basal application of the reduced N rate. Compared with CM treatment, RB treatment significantly decreased sweet potato yield by 10.7%, averaged over two years. Moreover, RB treatment recorded a 2.3% higher number of storage roots but an 8.0% lower mean storage root weight relative to that in the CM treatment, but the differences were not statistically significant. However, a split application of reduced N fertilizer could enhance the storage root number and mean storage root weight, resulting in a dramatically higher sweet potato yield relative to the basal fertilizer application method. In comparison with CM treatment, RS treatments resulted in significant increases in the average storage root number, mean storage root weight and sweet potato yield by 12.3%, 10.2% and 22.1%, respectively.

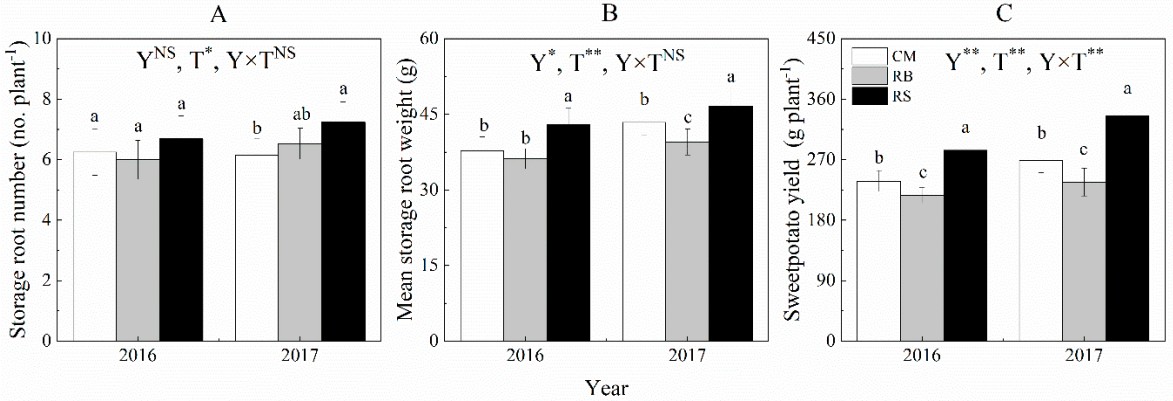

**Figure 2.** Storage root number (**A**), mean storage root weight (**B**) and sweet potato yield (**C**) under the three N management strategies in 2016 and 2017. Note: CM: conventional N management; RB: 20.0% reduced N was applied as basal; RS: split application of 20.0% reduced N at transplanting and 35 DAT. Y, year; T, treatment. Values are the treatment means for a given management strategy within a particular year (*n* = 3). Error bars represent the standard error of the mean. * indicate significance at *p* < 0.05 probability level. ** indicate significance at *p* < 0.01 probability level. NS indicate no significance. The different lowercase letters in the same column indicate significant differences at *p* ≤ 0.05.

### 3.3. N Content, C Content and C/N Ratio in Storage Roots of Sweet Potato

N management strategies had a significant effect on the N content, C content and C/N ratio in storage root (Figure 3). At 35 DAT, compared with CM treatment, RB and RS treatments significantly decreased the N content by 8.2% and 12.8% (averaged over two years) in the storage root, respectively. RB and RS treatments significantly elevated C content and C/N ratio by 5.5%, 12.2% and 13.8%, 27.8% relative to those of CM treatment, respectively. However, at 80 DAT, compared with CM treatment, the N content in storage root was reduced by 1.6% in the RB treatment, but this decrease was not significant. RS significantly increased the N content in storage root by 4.7% relative to CM treatment. The C content and C/N ratio of RB and RS treatments were −4.6%, 6.9% and −8.8%, 8.7% higher than those of CM treatment, respectively.

### 3.4. Endogenous Hormone Levels in Storage Root of Sweet Potato

At 35 DAT, CM treatment recorded the highest ABA content but the lowest IAA and Z + ZR contents in storage root. In contrast, RS treatment had the highest IAA and Z + ZR contents but the lowest ABA content in storage root (Figure 4). At 80 DAT, the ABA content in storage root showed a consistent trend for both years with treatments RS > CM > RB but exhibited inverse trends in the IAA and Z + ZR contents (Figure 4).

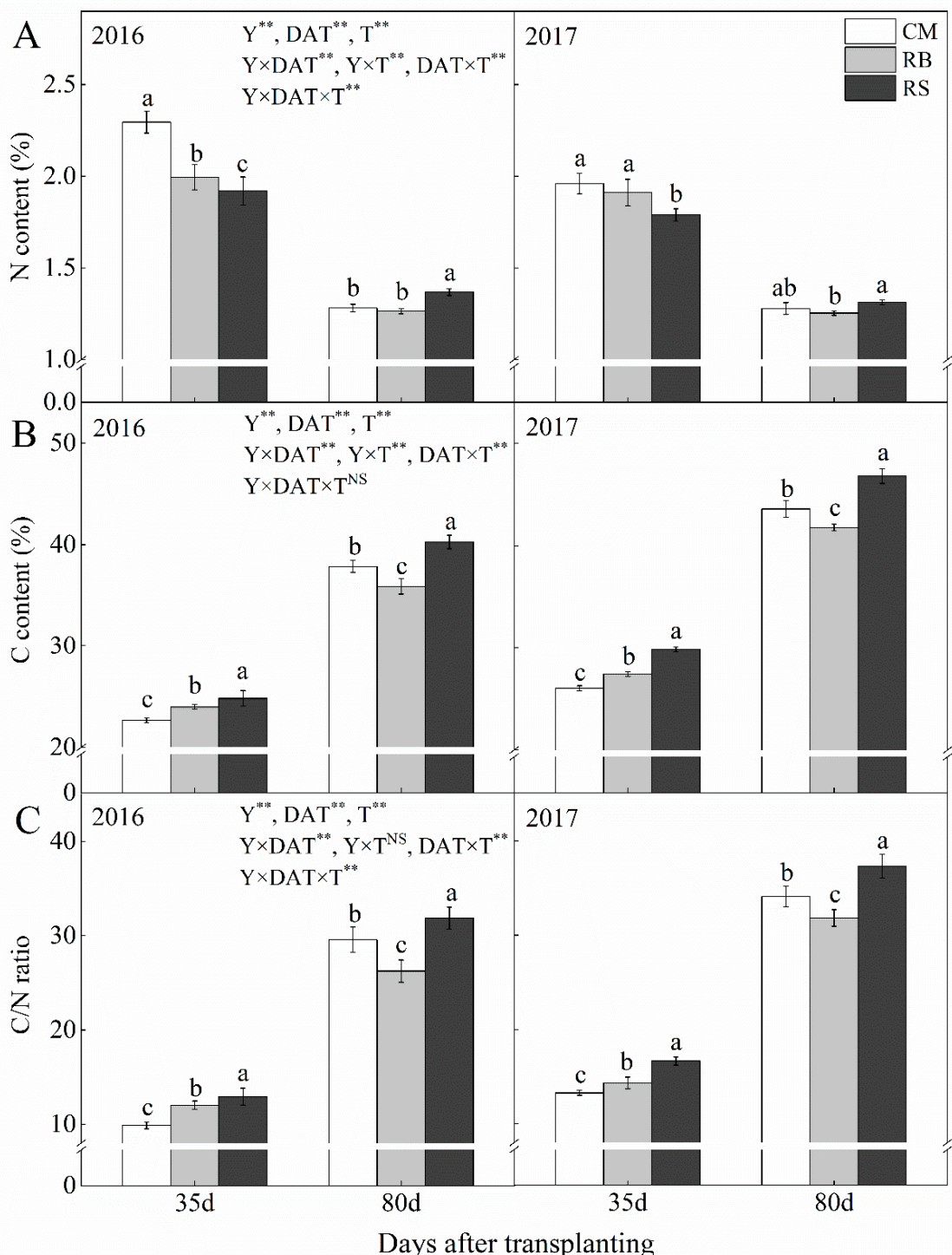

**Figure 3.** N content (**A**), C content (**B**) and C/N ratio (**C**) in the storage root in the three N management strategies at 35 and 80 DAT in 2016 and 2017. Note: CM: conventional N management; RB: 20.0% reduced N was applied as basal; RS: split application of 20.0% reduced N at transplanting and 35 DAT. Y, year; DAT, days after transplanting; T, treatment. Values are the treatment means for a given management strategy within a particular year ($n = 3$). Error bars represent the standard error of the mean. * indicate significance at $p < 0.05$ probability level. ** indicate significance at $p < 0.01$ probability level. [NS] indicate no significance. The different lowercase letters in the same column indicate significant differences at $p \leq 0.05$.

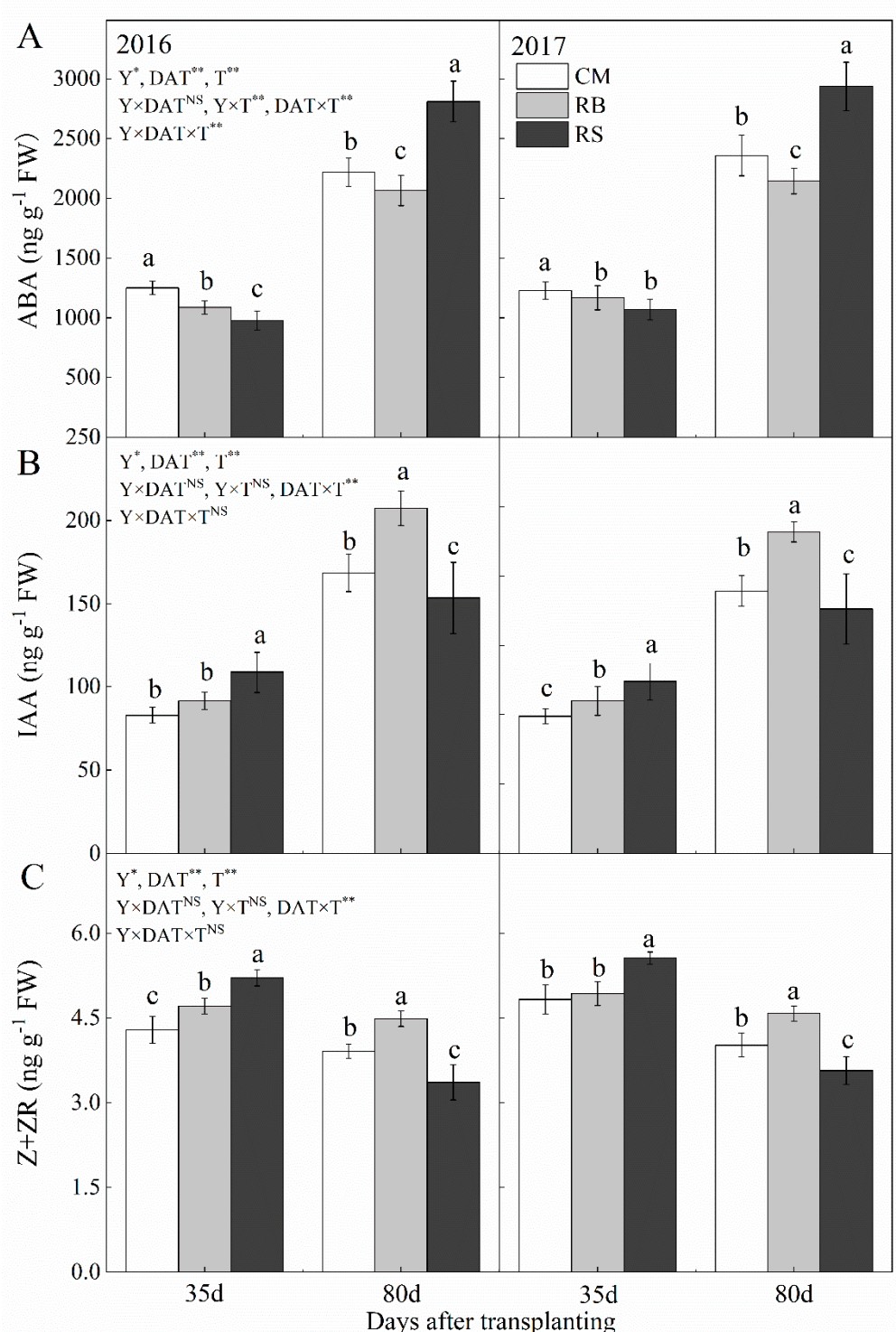

**Figure 4.** Abscisic acid (ABA, **A**), auxin (IAA, **B**) and zeatin and zeatin riboside (Z + ZR, **C**) levels in the storage root in the three N management strategies at 35 and 80 DAT in 2016 and 2017. Note: CM: conventional N management; RB: 20.0% reduced N was applied as basal; RS: split application of 20.0% reduced N at transplanting and 35 DAT. Y, year; DAT, days after transplanting; T, treatment. Values are the treatment means for a given management strategy within a particular year (*n* = 3). Error bars represent the standard error of the mean. * indicate significance at *p* < 0.05 probability level. ** indicate significance at *p* < 0.01 probability level. NS indicate no significance. The different lowercase letters in the same column indicate significant differences at *p* ≤ 0.05.

*3.5. Correlation between Endogenous Hormone Levels, N Content, C Content, C/N Ratio, Storage Root Number and Mean Storage Root Weight*

Correlation analysis demonstrated that the storage root number was positively and significantly correlated with the C content, C/N ratio, IAA content, Z + ZR content in storage root at 35 DAT but negatively and significantly correlated with the N and ABA contents in storage root at 35 DAT. In addition, the mean storage root weight was positively related to the N content, C content, C/N ratio and ABA content in storage root at 80 DAT (Table 1).

**Table 1.** Correlation coefficients between the storage root number, mean storage root weight, N content, C content, and C/N ratio and endogenous hormone levels at two different stages after transplanting for the three N management strategies in 2016 and 2017.

| Correlation with | N Content | | C Content | | C/N Ratio | | IAA | | ABA | | Z + ZR | |
|---|---|---|---|---|---|---|---|---|---|---|---|---|
| | 35 DAT | 80 DAT | 35 DAT | 80 DAT | 35 DAT | 80 DAT | 35 DAT | 80 DAT | 35 DAT | 80 DAT | 35 DAT | 80 DAT |
| Storage root number | −0.495 * | −0.006 | −0.526 * | 0.445 | 0.559 * | 0.516 * | 0.497 * | −0.175 | −0.537 * | 0.369 | 0.865 ** | 0.448 |
| Storage root weight | −0.554 * | 0.593 ** | −0.711 ** | 0.816 ** | 0.688 ** | 0.810 ** | 0.465 | −0.581 * | −0.668 ** | 0.866 ** | 0.112 | −0.432 |

Note: ABA; abscisic acid; DAT; days after transplanting; IAA; auxin; Z+ZR; zeatin and zeatin riboside. $n$ = 18, $R_{0.05}$ = 0.468, $R_{0.01}$ = 0.590. * Indicates significance of the correlation at the 0.05 level. ** Indicates significance of the correlation at the 0.01 level.

## 4. Discussion

Proper fertilization management strategies are critical to achieving higher sweet potato yields. In the present study, a split application with 20.0% reduced N fertilizer, which is a preferable agronomic practice, effectively increased sweet potato yield (Figure 2). Similar results were observed by Kim et al. [22], who found an increase in sweet potato yield by splitting the application of N at transplanting and 30 DAT. The split application of 20.0% reduced N at transplanting and 35 DAT resulted in a 22.1% higher storage root yield by increasing the storage root number by 12.3% and the mean storage root weight by 10.2%. It is also known that there is an interaction between N signaling, C-N metabolism and endogenous hormone activity. Therefore, we further analyzed the change characteristics of C-N metabolism and endogenous hormone during the storage root formation and bulking period and their relationships with storage root number and mean storage root weight.

*4.1. Enhanced IAA and Z + ZR Contents and Reduced ABA Content, Together with Increased C/N Ratio during the Storage Root Formation Period Promoted Storage Root Formation*

It has been reported that the growth period from 7 to 28 DAT is crucial in determining whether sweet potato adventitious root develop as storage roots or nonstorage roots [28]. The transplant-related and N management variables at transplanting could have a significant impact on early storage root formation and storage root number [29]. Sweet potato plants have a low demand for N during the storage root formation period, and excess N supply not only inhibits the cambium activity of adventitious root, which will in turn cause the lignification of stele cells and retard storage root formation [30,31]. In this study, RS treatment not only reduced the N application rate but also split only 50.0% of the total N at transplanting; thus, adverse effects of N supply on the number of storage roots formed could be minimized, resulting in the highest storage root number under RS treatment, which was in agreement with results from previous studies.

Sweet potato adventitious roots differentiate into fibrous root, pencil root or storage root in relation to the balance between the activity of the primary cambium and the lignification of the central stele [32]. IAA is involved in maintaining the meristematic state of cambial zone cells and is associated with lignification in storage root [33]. Z + ZR participates in the activation of the vascular cambium through concentrating around the root in the primary cambium and promoting cell division [34]. ABA inhibits the meristematic activity of apical cells and cell elongation, hence causing an aggravated degree of lignification in the column [13] and the proliferation of cambial cells that formed starch-accumulating parenchyma cells. It has been confirmed that higher levels of IAA in the root were obtained in a low

N condition compared with that in a high N condition [10,35]. ABA flows from root to stem caused by N deficiency led to a reduced ABA in the root [36]. Moreover, Z + ZR in the rice root decreased with increasing N application [37]. These results are consistent with the present study, indicating that a decreased N content in storage roots under RS treatment during the storage root formation period induced an increase in the IAA and Z + ZR contents but an decrease in the ABA content (Figure 4), which are conductive to promote cell division and expansion and reduce the degree of lignification [38,39]. In addition, promoting starch accumulation in the amyloid of parenchyma cells is also conductive to promoting adventitious root differentiation into storage roots [40]. According to Duan et al. [17], excess N supply led to shoot overgrowth and reduced the amount of assimilates availability for the storage root, and thus decreased the C/N ratio in the storage root. In our study, the lowest AV-N level under RS treatment reduced N uptake, but increased C content and C/N ratio in the storage root at 35 DAT (Figure 3B,C), which facilitated more starch synthesis and storage in the amyloid of parenchyma cells, and resulted in more storage root formation. Note that no difference in the storage root number was found between treatments in 2016, which may be due to the mutual restriction of yield components. Compared with CM treatment, we found that RS treatment increased the storage root number and mean storage root weight by 7.0% and 13.8% in 2016, but it increased the storage root number and mean storage root weight by 17.7% and 7.4% in 2017, respectively. Therefore, we speculate that RS treatment enhanced storage root formation during the storage root formation period, but some storage roots failed to expand during the storage root bulking period due to photosynthate competition between sweet potatoes. In general, these results indicate that splitting 50.0% of the total N at transplanting led to a lower AV-N condition and reduced N uptake, which induced increased contents of IAA and Z + ZR but decreased ABA content, together with elevated C content and C/N ratio in the storage root. These phenomena promoted more fibrous root to develop into storage root.

*4.2. Increased ABA Content Promoted C Allocated in Storage Root during the Storage Root Bulking Period and Increased the Mean Storage Root Weight*

The storage root weight is closely linked with storage root bulking. Storage root bulking is caused by the increase in the number of secondary xylem and phloem cells (mostly starch-storing parenchyma cells) that in turn is caused by anomalous cambia cell division and proliferation with substantial filling with starch [41,42]. In our study, RS treatment split 50.0% of the total N at 35 DAT and increased the AV-N content in soil by decreasing N waste during the storage root bulking period. Furthermore, our previous research found that RS treatment had the highest fibrous root number to access and uptake more N, resulting in an increased N content (Figure 3A), and leading to delayed leaf senescence and increased photosynthate during the storage root bulking period [43]. Moreover, RS treatment that recorded a higher ABA content (Figure 4A)—which not only activates cell division at the secondary meristem in the xylem [44,45] but also increases sink strength and promotes transportation and unloading of assimilates to starch-storing parenchyma cells by regulating carbon metabolism enzyme activity [46,47]—resulted in the highest C content and C/N ratio in the storage root during the storage root bulking period (Figure 3B,C). Therefore, RS treatment increased the mean storage root weight. Based on our findings, splitting 50.0% of the total N at 35 DAT could result in increased AV-N and elevated N uptake during the storage root bulking period, which increased the ABA content and promoted C allocation into the storage root, thereby increasing the mean storage root weight.

## 5. Conclusions

The split N application increased the sweet potato yield by increasing both the storage root number and mean storage root weight under a reduced N supply. The increased storage root number was attributed to the elevated contents of IAA and Z + ZR, and reduced ABA content, as well as enhanced C/N ratio in the storage root due to decreased N content in the storage root during the

storage root formation period; the higher ABA content due to the appropriately higher N content during the storage root bulking period promoted C allocation into the storage root, which led to an increased mean storage root weight. The changes in C-N metabolism and endogenous hormone levels with the split N application ensured proper storage root formation and storage root bulking, which are vital for obtaining high sweet potato yields under reduced N input.

**Author Contributions:** X.D. and L.K. conceived and designed the experiments; X.D., X.Z. and M.X. performed part of the experiments and analyzed the data; X.Z. prepared the manuscript; X.D., X.Z., M.X. and L.K. revised the manuscript. All authors have read and approved the final manuscript.

**Funding:** This work was supported by the National Natural Science Foundation of China under Grant No. 31601266.

**Conflicts of Interest:** The authors declare that there is no conflict of interest.

**Disclosure Statement:** No potential conflict of interest was reported by the authors.

## Abbreviations

| | |
|---|---|
| ABA | abscisic acid |
| AV-N | Soil-available N |
| CM | conventional N management |
| DAT | days after transplanting |
| IAA | auxin |
| RB | 20.0% reduced N was applied as basal |
| RS | split application of 20.0% reduced N at transplanting and 35 DAT |
| Z + ZR | zeatin and zeatin riboside |

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
