# Peer review of "Split Application under Reduced Nitrogen Rate Favors High Yield by Altering Endogenous Hormones and C/N Ratio in Sweet Potato"

_agronomy, doi:10.3390/agronomy10091243_

Round 1

Reviewer 1 Report

Please see attached comments 

Reviewer 2 Report

Congratulations on an excellent scientific contribution to sweet potato production in the field of agronomy. The experiment was very designed, appropriately conducted and relevant data collected and analyzed. Results were appropriately presented graphically as figures 1, 2, 3, 4, 5, and in Table 1, and adequately discussed. The study conclusions addressed the objectives of the study. Excellent scientific contribution. 

On line 90 where the caption of Figure 1 appears, please correct the caption as follows;

"Figure 1. Average temperature and precipitation at the experimental site in 2016 and 2017."

The caption appears in the manuscript as follows:

"Figure 1.Averagetemperatureandprecipitationattheexperimentalsitein2016 and2017."

Author Response

We would first like to thank you for your review of our manuscript. We feel that the comments were important for the improvement of our paper.

In response to comments from Reviewer 2:

Q: On line 90 where the caption of Figure 1 appears, please correct the caption as follows;

"Figure 1. Average temperature and precipitation at the experimental site in 2016 and 2017."

The caption appears in the manuscript as follows:

"Figure 1.Averagetemperatureandprecipitationattheexperimentalsitein2016 and2017."

A: We have deleted the Figure 1. We apologize and thank you for noticing this error.

Reviewer 3 Report

This article is well written and advances the knowledge into the plant mechanisms dealing with sweet potatoes yield under split timing and amount of N fertilization practice. 

I would like to encourage the authors to include the ANOVA table in the results section and improve the methodology description. Moreover, if there is some data on the leaf area, for example, it would be good to show it. Did the reduced N decrease the leaf area? This may, or may not, have implicates in other crop management practices such as row spacing and plant population. Can a split N application modify the actual management to also capture more solar radiation (and therefore more yield) or it is already maximized?

Here are my comments by lines. 

Abstract

#24-25  Consistency problem. AV-N is well defined but Z+ZR is not defined in the abstract. Then, both terms are in the abbreviation section.

Introduction

#37-38 Add a reference to this statement. Also, add why sweet potatoes are important (rank #7th). Is it because of their production?

Material and methods

#83-88 Needs more description. Usually, pot experiments are into greenhouses, however, this experiment seems to be in pots but in the field. If that is correct, it needs to be clarified and well stated.

#91-105 Needs to explicitly state the experimental design. Was it a complete randomized design? If there were 50 pots per treatment, how did authors organize the pots in the field? It needs more description of the distance between pots and the distribution of the treatments in the space. Did the authors use some pots as borders?

#167-183 Do the authors have any explanation of why the storage root number was not different between treatments in 2016?

Round 2

Reviewer 1 Report

Thank you for addressing all comments thoroughly.